# Experimental Investigation on Combustion and Performance of Diesel Engine under High Exhaust Back Pressure

**Li Huang** [1,2], **Junjie Liu** [3,*], **Rui Liu** [2,4], **Yang Wang** [3] **and Long Liu** [3]

1 College of Mechanical and Power Engineering, Shanghai Jiao Tong University, Shanghai 200240, China
2 Shanghai Marine Diesel Engine Research Institute, Shanghai 201111, China
3 College of Power and Energy Engineering, Harbin Engineering University, Harbin 150001, China
4 College of Power Engineering and Engineering Thermophysics, Shanghai Jiao Tong University, Shanghai 200240, China
* Correspondence: ljj1010069637@hrbeu.edu.cn; Tel.: +86-17390601250

**Abstract:** The use of exhaust gas recirculation, complex after-treatment systems, advanced technology of high-strength engines, and underwater exhaust will lead to increased diesel exhaust back pressure (EBP). This will increase the residual exhaust gas and the exchange temperature in the cylinder and reduce the fresh air charged in the next cycle. In this work, the effects of two high EBP conditions (10 kPa and 25 kPa) on the performance of medium-speed ship engines under different loads are explored through experiments. The results show that the increase in EBP from 10 kPa to 25 kPa has little effect on the heat release rate, engine power, and engine start-up time. However, it will lead to ignition advance and the maximum pressure rise rate, peak pressure, and exhaust temperature increase. The increase in EBP has a more significant impact on the small valve overlap angle. Because the reduction in the valve overlap angle has led to an increase in the residual exhaust gas, further increases in EBP causes residual exhaust gas effects to be more pronounced. The effect of increasing EBP on fuel consumption depends primarily on which effect of exhaust back pressure on temperature and fresh air intake dominates.

**Keywords:** diesel engine; exhaust back pressure; combustion characteristics; engine performance





## 1. Introduction

Diesel engines are widely used in industry, agriculture, transportation, etc. It is the main power source for agricultural machinery, engineering machinery, ships, locomotives, military vehicles, mobile and emergency electric stations, and so on [1]. In order to ensure the performance of diesel engines in special fields and environments, diesel engine technology has been continuously developed and strengthened. The high-pressure common rail system properly separates the functions of fuel injection metering, pressure rise, and fuel injection timing, which realizes a highly autonomous control of fuel injection and makes it easier to achieve high-pressure fuel injection [2]. Vera-Tudela et al. [3] indicated that very high injection pressures (up to 5000 bar) lead to better mixing and faster ignitions in diesel engines. Salehi et al. [4] conducted a large eddy simulation of high-pressure spray with injection pressure as the focus and studied the effect of injection pressure changes on the spray structure at 50 MPa, 100 MPa, and 150 MPa. The study found that an increase in injection pressure enhanced the vapor penetration length but had less effect on the liquid penetration length. Liu et al. [5] studied the application of gaseous ammonia and liquid ammonia in a low-speed two-stroke engine under the high-pressure injection condition of 200 MPa. A conceptual model for turbulent ignition in high-pressure spray flames shows that the low-temperature reactions occur at the radial spray periphery, then the high-temperature ignition occurs over a broad range of rich equivalence ratios, and finally, the suddenly forming steep gradients from successful high-temperature ignition initiates

the propagation of a turbulent flame [6,7]. It is an important method to strengthen the engine by increasing the intake air volume of the engine through the two-stage turbocharged technology to improve the power and economy of the engine. The reasonable matching of a two-stage turbocharging system can allow the system to operate at high efficiency at all speeds and under all load conditions. Studies have shown that the pressure ratio distribution should allow the high-pressure and low-pressure compressors to do most of the work at low and high speeds, respectively. In medium-speed conditions, the pressure ratio distribution should be 1:1 [8]. Zhu et al. [9] compared the important performance of three turbocharging methods of asymmetric twin-scroll, two-stage, and variable geometry turbocharging at different exhaust gas recirculation (EGR) rates. The results show that when the engine is fully loaded, the two-stage turbocharging performs best when the EGR rate is lower than 29%, and when the EGR rate is higher than 29%, asymmetric twin-scroll turbocharging is the best choice. Some scholars have studied the regulation law of the two-stage turbocharging system at different altitudes [10,11]. The intake pressure, peak pressure, and heat release rate of the two-stage turbocharging system diesel engine will be greatly reduced under high altitude conditions. Under the same high-altitude conditions, as the speed increases, the decrease in torque, fuel consumption, and fuel consumption rate will increase. The two-stage turbocharging system can effectively meet the overall efficiency at high altitude and low altitude. A variable compression ratio has a good effect on improving the fuel economy and emission performance of diesel engines. Therefore, research on variable compression ratios has been favored in recent years. Wittek [12] and Milojevi [13], respectively, studied the structural technology of realizing variable compression ratio and the influence of variable compression ratio on diesel engine fuel economy and emission characteristics. The value of the optimal compression ratio at which the engine runs with minimal fuel consumption increases with the increase in load, while particle emission is the smallest for medium loads and increases if the engine runs at low or high loads.

From the above research techniques, it can be seen that the research on diesel engines has been developing towards high intensity and high maneuverability. Blasio et al. [14] achieved a power density of 70 kw/L by developing a fuel injection system with an injection pressure capable of exceeding 2500 bar. Qin et al. [15] studied the degree of ablation of different piston materials and the main reasons for the ablation in highly strengthened engines. The research results of Gugulothu show that compared with the hemispherical combustion chamber, the toroidal combustion chamber improves the air swirl and the brake thermal efficiency and reduces the CO and CH emissions, which is more suitable for high-strengthened diesel engines [16]. Studies found that fuzzy-PID and sliding mode control strategies have a significant effect on improving the response speed of the diesel engine speed control system [17,18]. Liu et al. [19] obtained good comprehensive performance of diesel fuel injection systems by developing a new permanent magnet, in which the driving response speed was improved by 11.9%.

In recent years, laws and regulations have strictly restricted the emission of marine engines, which has prompted the widespread application of EGR and after-treatment systems. EGR also has a significant effect on alternative fuel engines with new combustion modes [20]. However, the EGR and after-treatment systems, the advanced technology of high-intensity engines, and the underwater exhaust of cargo ship engines will all lead to an increase in exhaust back pressure (EBP) [21–23]. In addition, the complexity of the after-treatment systems and the difference in the draft of the cargo ships make the variation of the EBP inconsistent. EBP has a profound impact on the combustion characteristics and performance parameters of an engine, so EBP has been extensively studied in recent years. Through simulation, Sapra et al. [22] found that, compared with the large valve overlap and constant pressure turbocharged engines, the small valve overlap and pulse turbocharged engines can better handle the high exhaust pressure. The simulation results of Tauzia et al. show that the effects of dynamic back pressure and static back pressure on engine performance show consistencies, which are all manifested as a decrease in the air–fuel ratio, an increase in exhaust temperature, and a decrease in the pressure difference

between the inlet and outlet of the cylinder [23]. Sapra found that back pressure fluctuations have almost no effect on engine performance at high engine rpms, while the effects are significant at low rpms [24]. Due to the high cost and complex experimental equipment of diesel engine tests, although there are some related tests of EBP, they mainly focus on single-cylinder engines and gasoline engines, which are not suitable for marine applications. Jardder et al. [25] compared the brake thermal efficiency, NOx, CO, and exhaust gas temperature, and it is considered that the increase in EBP is harmless to the CI engine when the EBP is less than 40 mm of Hg. Mittal et al. [26] mainly studied the effects of high EBP and low EBP on the emissions of NOx, Soot, PM, CO, and total HC under different loads. The emissions results showed that, under high EBP, the concentration of NOx was reduced. Soot, PM, and CO were increased, but no significant difference was observed in HC emission.

In this work, the performance of high EBP caused by three different practical application conditions under different loads of a medium-speed marine engine was discussed from the perspective of experiments while ensuring the optimal power of the whole engine, and the results are different from the traditional single-cylinder test results under single-variable conditions. This research is of great significance for expanding the applicability of the engine on ships and also provides guidance for the performance improvement of the engine.

## 2. Methods

### 2.1. Experimental Setup

A schematic of the experimental setup is shown in Figure 1. The experiments were conducted on a 16-cylinder turbocharged marine diesel engine, which was equipped with a common rail fuel injection system, lubricating oil system, coolant system, starting system, and intake/exhaust system. The engine had a bore diameter of 270 mm, a stroke length of 330 mm, and a speed of 1000 r/min. Its common rail pressure can reach 1600 bar, and the load is controlled by adjusting the fuel injection quantity. The diesel engine specifications are given in Table 1. The purpose of controlling the EBP is achieved by controlling the opening of the flow control valve on the exhaust manifold. A fuel flow meter (Emerson) was equipped before the common rail. Additionally, the unit was equipped with thermocouples (Trafag) and pressure sensors (CMR) at the outlet of the air compressor, intake manifold inlet, exhaust manifold outlet, diesel fuel line and the cylinder, etc., to obtain the temperature and pressure of various locations of the unit to evaluate the engine performance.

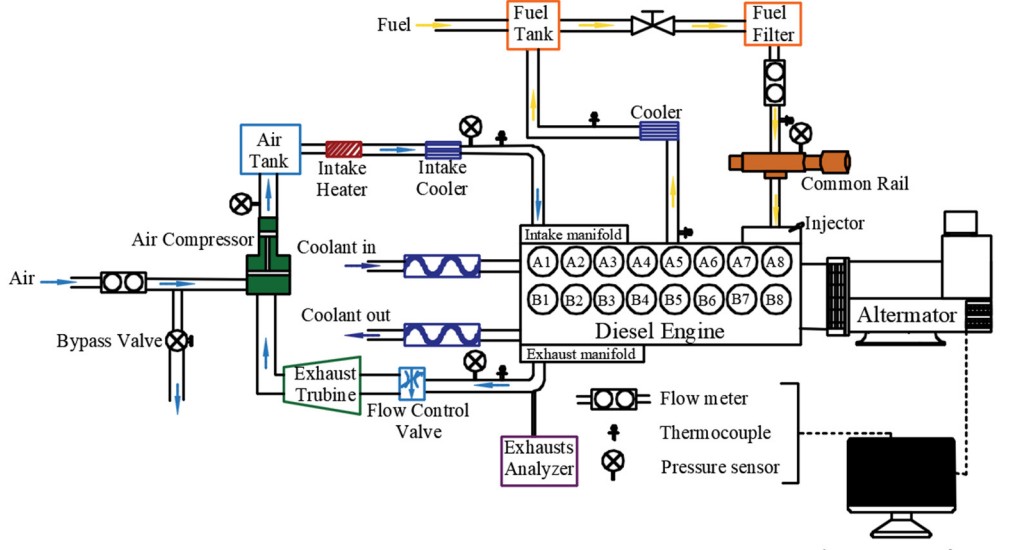

**Figure 1.** Schematic of the experimental setup.

**Table 1.** Diesel engine specifications.

| Parameters | Value |
|---|---|
| Bore (mm) | 270 |
| Stroke (mm) | 330 |
| Connecting rod length (mm) | 660 |
| Number of nozzle holes | 10 |
| Spray angle (°) | 148 |
| Speed (r/min) | 1000 |

*2.2. Experimental Procedure*

The effect of injector geometry and fuel injection parameters on combustion characteristics and performance parameters were investigated by varying the nozzle diameters, injection pressures, and the intake and exhaust parameters at different engine loads. The air-cooled resistance box is used to absorb the electrical load generated by the generator. The commonly used loads of 50%, 75%, and 100% of generators were selected as the research loads. By adjusting the resistance value of the resistance box to determine and change the load. Seawater cooling was used in the experiment. Under 100% load, the inlet temperature and the pressure of the coolant were maintained at 70 °C and 3.8 bar, while the temperatures of the 50% load and 75% load were slightly higher. At 100% load, the fuel inlet temperature and pressure were maintained at 37.5 °C and 5.7 bar, and the inlet temperature and pressure of lubricating oil were maintained at 65.4 °C and 4.8 bar. The temperature and pressure at 50% load and 75% load are not much different from those at 100% load. The experiments were performed for three cases, which did not pursue a single variable but ensured the optimal power of the whole machine. The parameter details are given in Table 2. Case 1 and Case 3 were tested on the conditions of 10 kPa and 25 kPa EBP, respectively. In order to reduce the air exchange loss and gas backflow, Case 3 adjusted the timing of the intake valve opening and the exhaust valve closing to reduce the valve overlap angle. The intake/exhaust valve timing of the three cases is illustrated in Figure 2.

**Table 2.** Parameter details of the three cases.

| Parameters | Load | Case 1 | Case 2 | Case 3 |
|---|---|---|---|---|
| Nozzle diameter (mm) | 50%<br>75%<br>100% | 0.475 | 0.46 | 0.46 |
| Rail pressure (bar) | 50%<br>75%<br>100% | 1600 | 1500 | 1600 |
| EBP (kPa) | 50%<br>75%<br>100% | 10/25 | 10 | 10/25 |
| Intake pressure (bar) | 50%<br>75%<br>100% | 1.7/1.5<br>2.7/2.3<br>3.5/3.1 | 1.1<br>1.8<br>2.5 | 1.3/1.2<br>2.1/1.8<br>3.0/2.5 |
| Injection timing (° CA BTDC) | 50%<br>75%<br>100% | 6/7<br>4/8<br>3/9 | 8<br>7<br>7 | 9/9<br>8/8<br>6/9 |
| lambda | 50%<br>75%<br>100% | 2.6/2.6<br>2.6/2.4<br>2.3/2.3 | 2.84<br>2.75<br>2.6 | 2.43/2.32<br>2.35/2.2<br>2.32/2.07 |

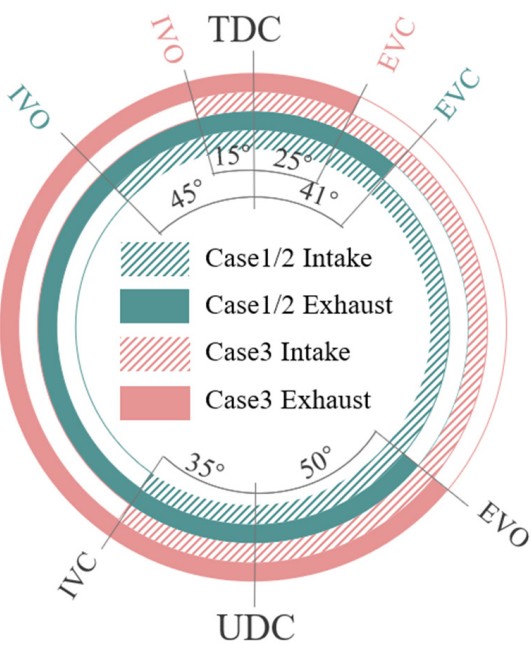

**Figure 2.** Intake/Exhaust valve timing of the three cases.

## 3. Results and Discussion

In this section, the effects of tested conditions on the parameters studied are discussed, and the results of the experiment are given separately in the sub-sections on combustion characteristics, engine performance, and comprehensive comparison.

### 3.1. Combustion Characteristics

Figure 3 presents the in-cylinder pressure and heat release rate (HRR) of Case 1, Case 2, and Case 3 at three engine loads of 50%, 75%, and 100%, and an EBP of 10 kPa and 25 kPa. When the EBP is 10 kPa, as the load increases, the intake air and the fuel injection increase, and the HRR is higher, but the fuel injection timing is delayed, so the combustion timing is delayed. Corresponding to the injection timing, the HRR of Case 1 rises later than that of Case 2 and Case 3, but the intake air of Case 1 increases, the combustion in the cylinder is more intense, and the HRR is higher. Because the fuel injection pressure of Case 2 is lower, the spray penetration distance is smaller, so the atomization is worse than that of Case 1 and Case 3, and the combustion HRR is lower.

At the 25 kPa EBP, with the increase in the load, the law of the heat release is consistent with the 10 kPa EBP condition. Despite the fact that the valve overlap angle of Case 3 is smaller, the fresh air in the cylinder is diluted, but the short exhaust time increases the initial temperature in the cylinder. When the injection timings of Case 3 and Case 1 are similar, the heat release rate of Case 3 rises earlier. This indicates that the thermal effect of the short exhaust time has a greater impact on the ignition time than the dilution effect of fresh air. At the same load and EBP, the peak heat release rate of Case 1 is always higher than that of Case 2 and Case 3, and it can be seen from Figure 4 that at the same load and EBP, the fuel consumption rates of the three cases are similar. This means that the start of combustion is mainly controlled by the fuel injection timing, and the initial temperature in the cylinder also has a relatively obvious influence, while the combustion rate is mainly controlled by the amount of intake air and the fuel–air mixing.

The CA5 (the crank angle at 5% cumulative heat release) and CA50 (the crank angle at 50% cumulative heat release) are the key parameters influencing the ignition and combustion of the engines. Figure 5 illustrates the CA5 and CA50 at different loads and EBPs. At 10 kPa EBP, CA5 is almost unchanged in Case 1 on different loads. Both Case 2 and Case 3 have ignition delay with the load increases, which is controlled by the injection timing.

However, due to the small overlap angle of the valve in Case 3, there is more residual exhaust gas, and the initial temperature in the cylinder is high, resulting in a significantly earlier ignition time for Case 3. At 25 kPa EBP, the increase in EBP also leads to an increase in residual exhaust gas [21], which makes the temperature in the cylinder higher than that at 10 kPa, but the ignition timing of Case 3 at 50% and 75% load is almost the same as that at 10 kPa, and on 100% load, the fuel injection time of Case 1 is advanced, but the ignition is later. This indicates that the CA5 is affected by many factors, such as in-cylinder initial temperature, fuel injection timing, and fuel–air mixing.

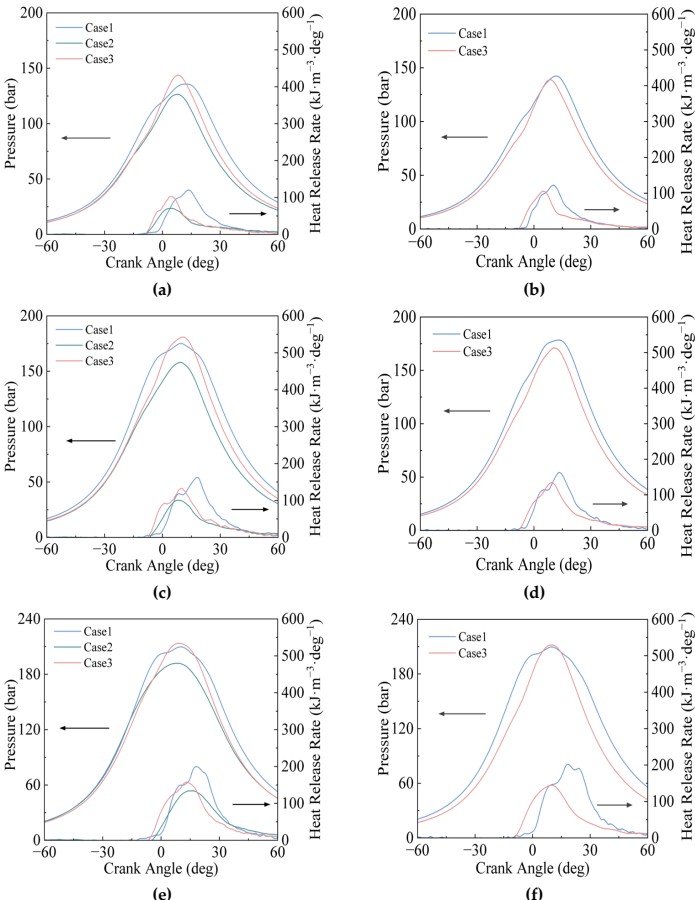

**Figure 3.** Combustion characteristics for variable EBP at different loads. (**a**) EBP = 10 kPa, load = 50%; (**b**) EBP = 25 kPa, load = 50%; (**c**) EBP = 10 kPa, load = 75%; (**d**) EBP = 25 kPa, load = 75%; (**e**) EBP = 10 kPa, load = 100%; (**f**) EBP = 25 kPa, load = 100%.

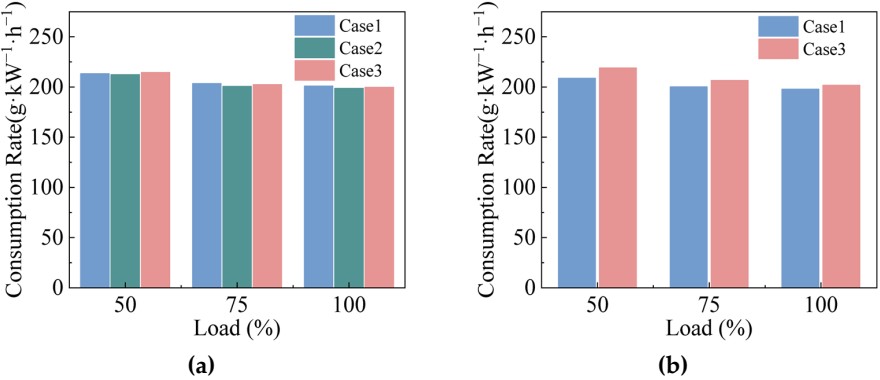

**Figure 4.** The trend of consumption rate for variable EBP at different loads. (**a**) EBP = 10 kPa; (**b**) EBP = 25 kPa.

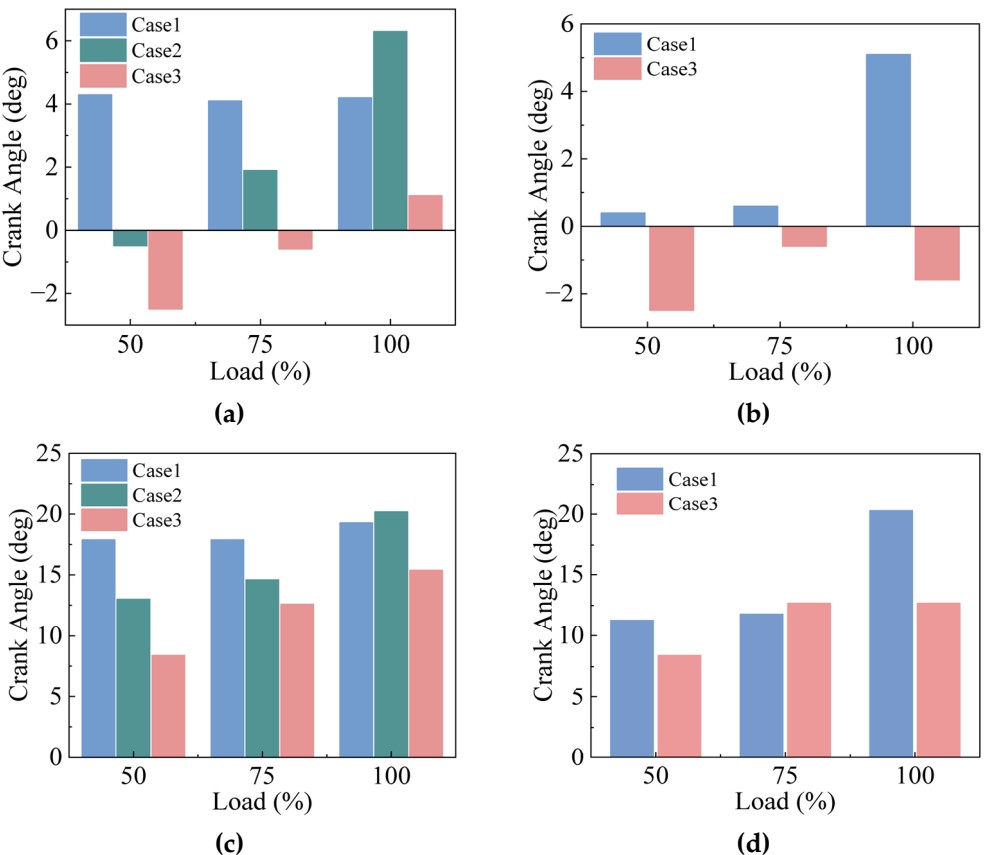

**Figure 5.** CA5/CA50 for variable EBP at different loads. (**a**) EBP = 10 kPa, CA5; (**b**) EBP = 25 kPa, CA5; (**c**) EBP = 10 kPa, CA50; (**d**) EBP = 25 kPa, CA50.

As shown in Figure 5, although the overall trend of CA50 changes with load variety is the same as that of CA5, the difference of CA50 of Case 2 on the three loads at 10 kPa EBP is smaller than that of CA5, and at the condition of 25 kPa, the difference of CA50 of Case 1 and Case 3 is also greater than that of CA5. It is seen that the trends of CA5 and CA50 are not exactly the same with the load increases. The CA5 that reflects the ignition is mainly controlled by the injection timing and premixed combustion, while the CA50 is mainly determined by the diffusion combustion.

The combustion duration is characterized by CA90-CA5, as shown in Figure 6. On 10 kPa EBP, at the same load, the combustion duration of three cases accord with the law: Case 2 > Case 3 > Case 1, which indicates that the combustion duration is affected by both ignition timing and the fuel–air mixing. Figure 6 shows that the CA90-CA5 duration is long because the partial wall-coated fuel evaporative mixing is not sufficient, resulting in the formation of a fuel–air mixture at a later time, delaying the combustion heat release and prolonging the combustion duration of Case 2 and Case 3. However, the fuel injection pressure and in-cylinder temperature of Case 3 are high, and the mixing conditions of fuel and air in the cylinder are better, so the combustion duration of Case 3 is shorter than that of Case 2. Especially with the increase in load, the CA50 of Case 2 and Case 3 increases, while CA90-CA5 decreases, which indicates that on low load, the after-combustion period of Case 2 and Case 3 is long. This is because Case 2 and Case 3 have low intake pressure at light load, less extra air, and poor fuel–air mixture in the later stage of combustion, so the afterburning phenomenon is serious, even reaching more than half of the entire combustion duration. The trend of the combustion duration of Case 1 and Case 3 is roughly the same at 25 kPa and 10 kPa EBP. Additionally, when the EBP increases at the same load, the change in the combustion duration is not obvious, which shows that the increase of 15 kPa EBP has no significant effect on the combustion duration within a reasonable range.

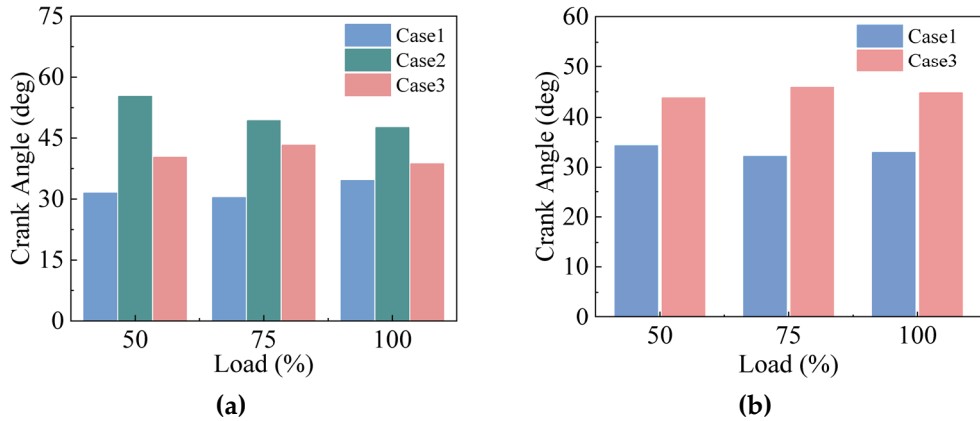

**Figure 6.** CA90-CA5 for variable EBP at different loads. (**a**) EBP = 10 kPa; (**b**) EBP = 25 kPa.

### 3.2. Engine Performance

Diesel engine performance can be evaluated in terms of start-up time, engine power, fuel consumption rate, peak pressure, pressure rise rate, and exhaust gas temperature. Figure 7 illustrates the start-up time of the engine for different cases at 10 kPa and 25 kPa. It can be seen that, at the condition of 10 kPa EBP, the time of engine start-up is Case 1 > Case 2 > Case 3. Compared with Case 1, the start-up time of Case 2 and Case 3 is shortened by 7.5% and 15.8%, respectively. Case 2 and Case 3 have smaller nozzle diameters, so the atomization conditions in the cylinder are better, and the engine start-up is faster. The injection timing of Case 3 is the earliest, so the ignition is faster, and the starting time is further accelerated than that of Case 2. At the condition of 25 kPa EBP, the engine start-up time of Case 1 is the same as that of 10 kPa EBP. In order to further study the improvement space of the start-up time, the single motor of Case 3 was replaced with dual motors in this experiment. As a result, at the condition of 25 kPa EBP, the engine start-up time of Case 3 was shortened by 28%.

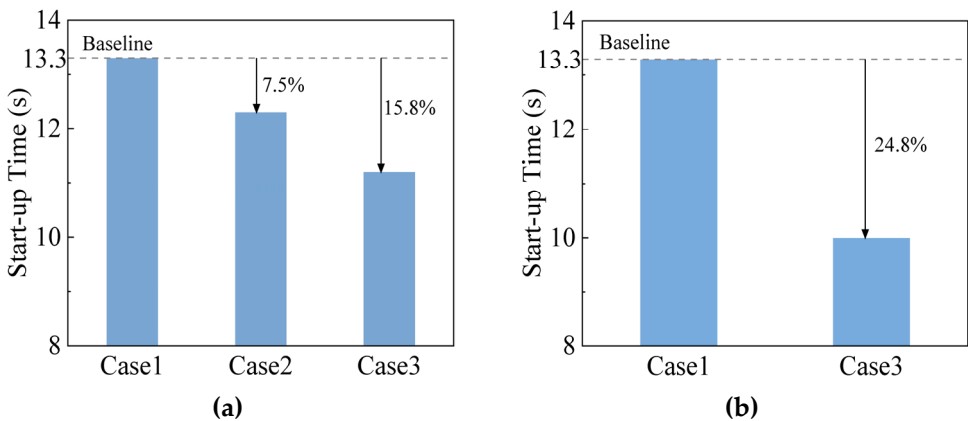

**Figure 7.** Start-up time for variable EBP at different cases. (**a**) EBP = 10 kPa; (**b**) EBP = 25 kPa.

The variation of power under different engine loads and EBPs for the tested cases are presented in Figure 8. As the engine load increases from 50% to 100%, the quality of fuel atomization continues to improve, so as the load increases, the power of different EBPs and different cases continue to increase. At different EBPs and engine loads, the power of Case 1 is higher than that of Case 3. The valve overlap angle and intake pressure of Case 1 are larger, the residual exhaust gas is less so the fuel and air mixture of Case 1 is better, and the energy release in the combustion process is higher. Especially when the engine is fully loaded, the power of Case 1 is significantly higher than that of Case 3. At 10 kPa EBP, the power of Case 3 is slightly higher than that of Case 2. Although the valve overlap angle of

Case 3 is smaller than that of Case 2, the intake pressure and injection pressure of Case 3 are higher. The high injection pressure increases the fuel injection penetration distance and improves the fuel–air mixing. Under different engine loads, Case 1 and Case 3 have almost no difference in engine power at 10 kPa and 25 kPa. Although the increase in EBP leads to the increase in residual exhaust gas, it also leads to the increase in the temperature of the cylinder. The two roles have almost the same effect on the fuel–air mixing in the cylinder, so the increase in EBP from 10 kPa to 25 kPa has little effect on engine power.

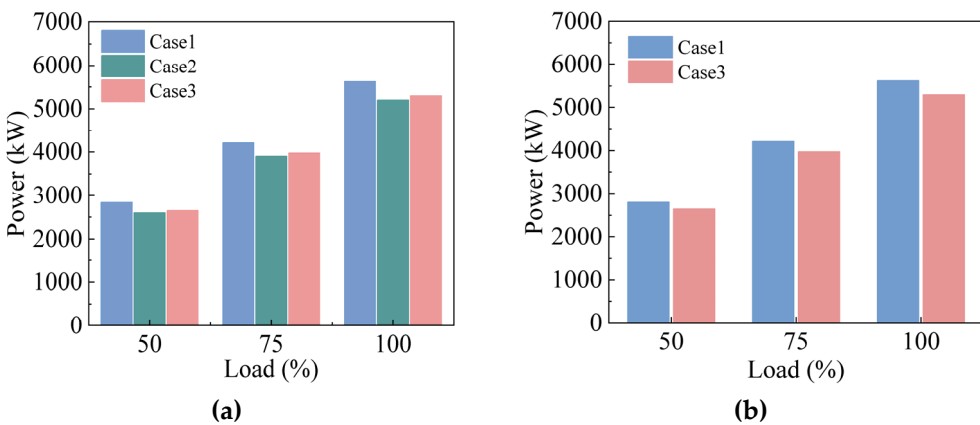

**Figure 8.** The trend of power for variable EBP at different loads. (**a**) EBP = 10 kPa; (**b**) EBP = 25 kPa.

Figure 9 illustrates the variation of peak pressure for three cases at different EBPs and loads. It can be observed that at the condition of 10 kPa EBP, with the increase in load, the fuel injection and intake air in the cylinder increase, which increases the combustion heat release, and then the in-cylinder peak pressure of the three cases increases significantly. The peak pressures of all the loads are ranked in the order of Case 3 > Case 1 > Case 2. It can be seen from Figure 5 that the ignition times of Case 3 at 50% load and 75% load are early (less than 0° CA), so the heat is released before the piston moves downward. In Case 1, although there is more fresh air in the cylinder and better mixing conditions of fuel and air, due to the long premixing time, the piston has already started to move downward when the heat is released, and the peak pressure of Case 1 is lower than that of Case 3 at three loads, as the CA5 discrepancy of the two cases decreases, the peak pressure discrepancy also decreases. This indicates that the thermal effect of early fire time has a greater impact on the peak pressure than the fuel–air mixing effect. The fuel injection pressure and intake pressure of Case 2 are lower than that of Case 1, leading to low heat release, so the peak pressure of Case 2 is the smallest.

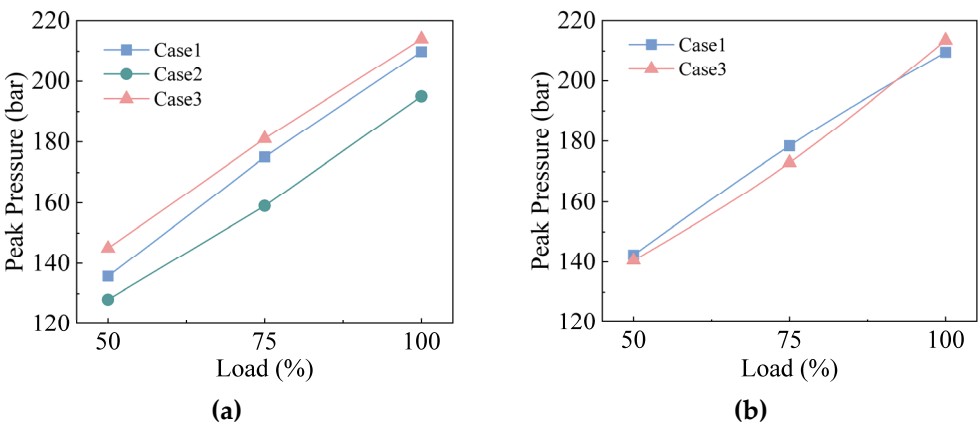

**Figure 9.** The trend of peak pressure for variable EBP at different loads. (**a**) EBP = 10 kPa; (**b**) EBP = 25 kPa.

At 25 kPa EBP, on 50% and 75% load, the peak pressure of Case 1 is higher than that of Case 3, while on 100% load, the peak pressure of Case 1 is lower than that of Case 3. Combined with Figure 5, it can be seen that at 100% load condition, the ignition lag of Case 1 makes the heat release concentrated in the expansion stroke. On 50% and 75% load, the peak pressure of Case 1 at the condition of 25 kPa EBP is higher than that at 10 kPa EBP condition, but on 100% load, it is similar in both EBPs. This shows that the peak pressure is mainly affected by the relative moment of ignition and 0° CA. Contrary to Case 1, on 50% and 75% load, the peak pressure of Case 3 at 25 kPa EBP is lower than that at 10 kPa EBP, while on 100% load, the peak pressure of Case 3 is also similar in both EBPs. This indicates that in addition to the ignition time, the increase in residual exhaust gas caused by the increase in EBP has a significant impact on the peak pressure at the condition of a small valve overlap angle.

Figure 4 illustrates the trend of fuel consumption rate with the variation of load and EBP. It is seen that on the condition of 10 kPa EBP, the fuel consumption rate of the three cases decreases with the increase in load. Mainly because the engine power is significantly lower at light load, as shown in Figure 8. As the injection timing advances, Case 3 > Case 2 > Case 1, the combustion start is advanced, the ignition delay period is prolonged, the fuel evaporation increases during the delay period, the more combustible mixture is formed, and the combustible mixture burns rapidly during the burning period, CA50 is significantly advanced (Figure 5). Due to the increase in heat release near the TDC, the thermal efficiency is increased, and the fuel consumption rate of the engine is reduced. However, in the premixed combustion period, the larger the amount of homogeneous mixture, the faster the combustion reaction, which is carried out when the cylinder volume is small when the piston is close to the TDC, resulting in a high maximum pressure rise rate, as shown in Figure 10. As the intake pressure increases, Case 1 > Case 3 > Case 2, the intake air volume increases, and the fuel–air mixing effect is better during the ignition delay period. As shown in Figure 4, at the same load, the fuel consumption rates of the three cases are similar. This indicates that the advance in injection timing and the sufficient intake of air have the same influence on the premixing period of the diesel engine. At 25 kPa EBP, the injection timing of Case 3 is the same as or slightly ahead of Case 1, but the valve overlap angle of Case 3 is small, resulting in insufficient fuel–air mixing during the ignition delay period, which increased fuel consumption. At the same load, the increase in EBP leads to a slight decrease in the fuel consumption rate of Case 1 and a slight increase in the fuel consumption rate of Case 3. This indicates that the residual exhaust gas in the cylinder caused by the increase in EBP is the main factor affecting the fuel combustion rate of Case 3 with a small valve overlap angle; while the high in-cylinder initial temperature caused by the increase in EBP further reduces the fuel consumption rate of Case 1.

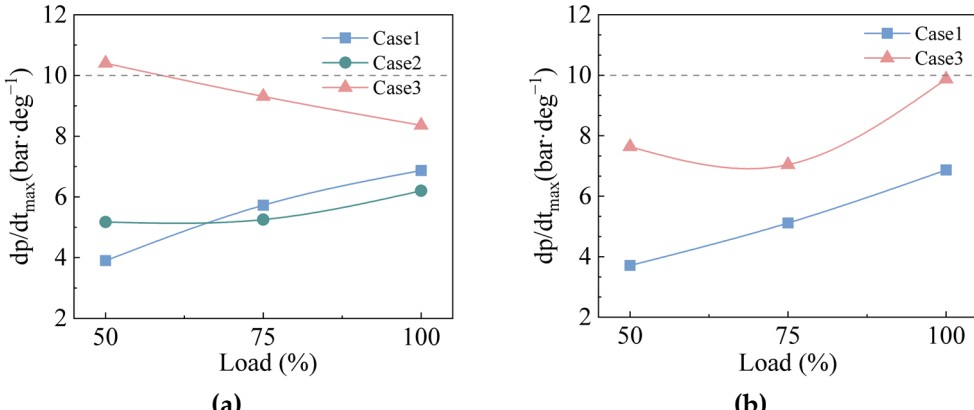

**Figure 10.** The trend of maximum pressure rise rate for variable EBP at different loads. (**a**) EBP = 10 kPa; (**b**) EBP = 25 kPa.

The pressure rise rate determines the smoothness of the diesel engine operation. Figure 10 illustrates the trend of the maximum pressure rise rate at different loads and EBP. The pressure rise rate is mainly affected by the mixture of fuel and air during the ignition delay period. Therefore, in addition to the injection timing, factors such as injection pressure, valve overlap angle, and intake air amount that affect the fuel–air mixing at the early stage of combustion will affect the pressure rise rate. At the two EBP conditions, although the injection timing of Case 3 is the same at 50% and 75% loads, it is affected by the high EBP and the low intake pressure. As a result, at the 25 kPa EBP of Case 3, the premixing situation during the ignition delay period is poor, and the pressure rise rate is low. In addition, the maximum pressure rise rate of Case 3 is significantly higher than that of Case 1 and Case 2 at the two EBP conditions. This indicates that the high initial temperature caused by the small valve overlap angle plays an important role in the homogeneity of the fuel and air mixing in the initial combustion.

Figure 11 illustrates the trend of exhaust temperature with the variation in EBP and load. As shown, at 10 kPa and 25 kPa EBP, the exhaust temperature of the three cases increases with the increase in load. This is mainly affected by the in-cylinder heat release rate, which increases with increasing load (Figure 3), as does the exhaust temperature. The exhaust temperature is controlled by the residual exhaust gas ratio [27]. At 10 kPa EBP, the opening and closing timings of the intake and exhaust valves of Case 1 and Case 2 are the same. Although the high intake pressure of Case 1 makes the exhaust rate fast under the same EBP, the high intake pressure also causes a large intake volume, and there is more exhaust gas at the initial time of exhaust. As shown in Figure 11, the exhaust temperature of Case 1 and Case 2 are almost the same on 50% load, but on 75% and 100% loads, the exhaust temperature of Case 1 is higher than that of Case 2, and as the load increases, the difference in exhaust temperature between Case 1 and Case 2 increases with the difference in intake pressure increases. This shows that the increase in intake pressure has a greater effect on the residual exhaust gas volume than on the exhaust gas rate, resulting in an increase in the exhaust gas temperature. The exhaust temperature of Case 3 is relatively higher than Case 1 and Case 2 at 10 kPa and 25 kPa EBPs. The main reason is that the exhaust overlap angle is small and the exhaust time is short, so the exhaust volume is small, and the residual exhaust gas in the cylinder is large. At the EBP of 25 kPa, the exhaust temperature of both Case 1 and Case 3 is higher than the exhaust temperature of the 10 kPa EBP. The increase in exhaust back pressure increases the exhaust resistance. In addition, the intake pressure at 25 kPa EBP is also slightly lower than the intake pressure of 10 kPa EBP. The double effect leads to the increase in the residual exhaust gas and the rise in exhaust temperature.

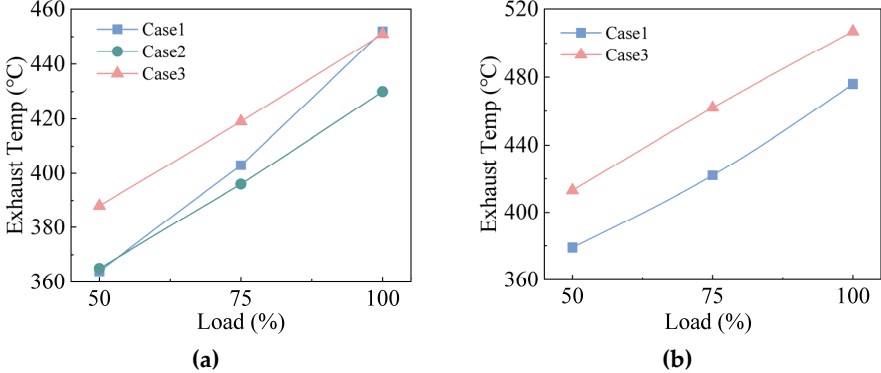

**Figure 11.** The trend of exhaust temperature for variable EBP at different loads. (**a**) EBP = 10 kPa; (**b**) EBP = 25 kPa.

### 3.3. Comprehensive Comparison

Figure 12 comprehensively compares engine power, consumption rate, and maximum pressure rise rate. At 10 kPa EBP, the trend of engine power conforms to Case 1 > Case 3 >

Case 2, the trend of maximum pressure rise rate conforms to Case 2 < Case 1 < Case 3, and the trend of consumption rate conforms to Case 2 < Case 3 < Case 1, expect at 50% load, the maximum pressure rise rates and consumption rates of Case 1 and Case 2 are opposite to the above situation. In addition, at 75% load, the power of Case 3 is reduced by 5.6% compared with Case 1, while the maximum pressure rise rate is increased by 62.5%. At 25 kPa EBP, under the three loads, the power trends all satisfy Case 1 > Case 3, the maximum pressure rise rate trends all satisfy Case 1 < Case 3, and the fuel consumption rates all satisfy Case 1 < Case 3. This shows that at high EBP, compared with Case 1, the engine performance of Case 3 is obviously inferior, but it is worth noting that Case 3 can meet some specific working conditions of the diesel generator.

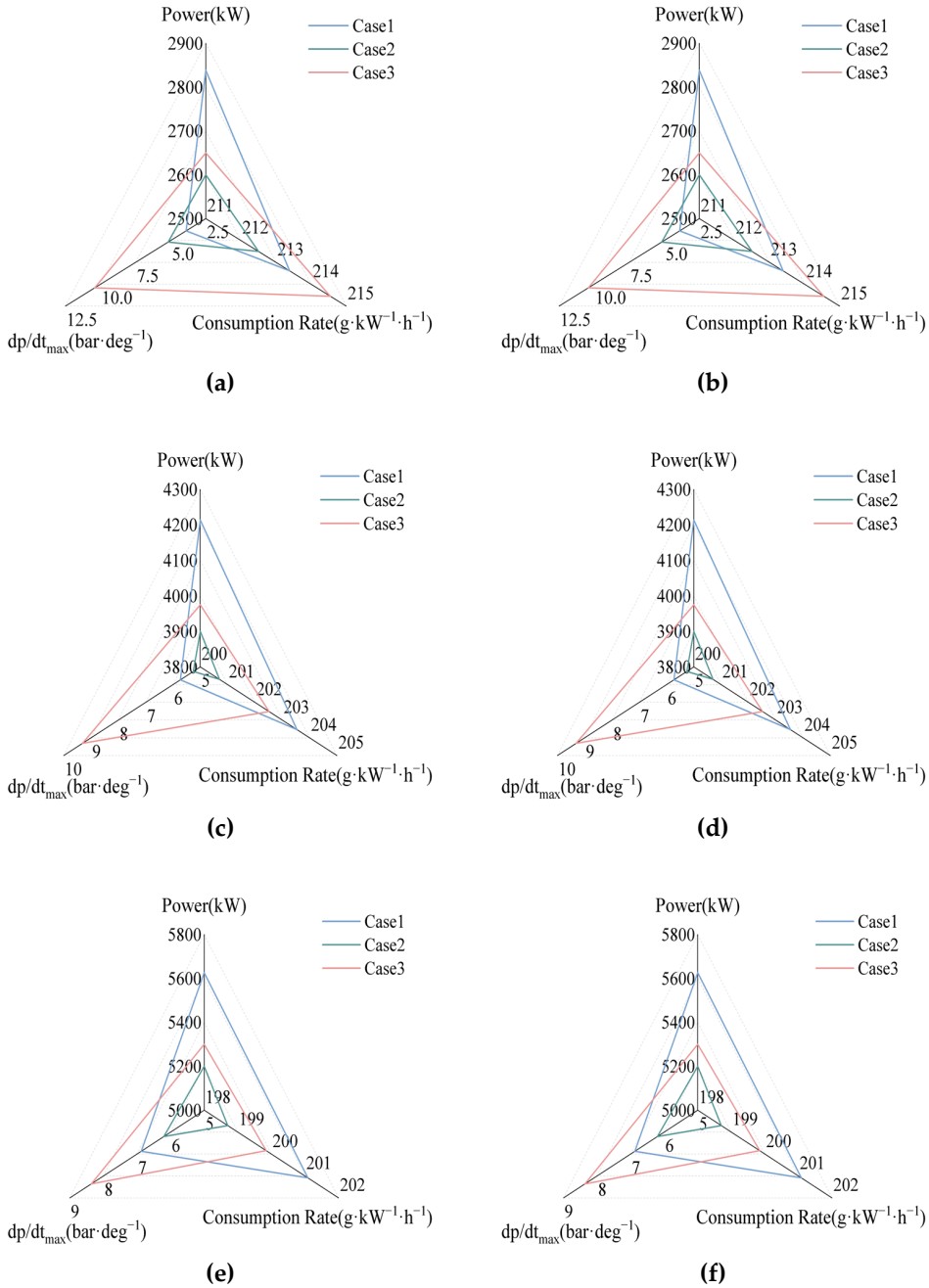

**Figure 12.** Comprehensive comparison for variable EBP at different loads. (**a**) EBP = 10 kPa, load = 50%; (**b**) EBP = 25 kPa, load = 50%; (**c**) EBP = 10 kPa, load = 75%; (**d**) EBP = 25 kPa, load = 75%; (**e**) EBP = 10 kPa, load = 100%; (**f**) EBP = 25 kPa, load = 100%.

According to the comprehensive comparison results, at high EBPs, cases can be selected according to the different emphases of the diesel generator sets. If the injection pressure and intensification of the diesel engine are limited by technology and materials, Case 2 can be selected, because of its low injection pressure, low maximum pressure rise rate, and low fuel consumption rate. Furthermore, the start-up time of Case 2 is faster (Figure 7). If the underwater exhaust of the ship causes high EBP and the diesel engine has a high level of intensity, Case 3 can be selected. Its valve overlap is small, which can effectively prevent gas backflow and seawater backflow, meaning that it has the best adaptability to underwater exhaust. In addition, Case 3 has the fastest start-up time. If the EGR and after-treatment system cause the increase in EBP, and the diesel engine does not have a high requirement on the start-up time, Case 1 can be selected, which has great advantages in power, maximum pressure rise rate, and fuel consumption rate.

## 4. Conclusions

In this work, three cases are formulated and applied to a medium-speed 16-cylinder diesel engine under the condition of high EBP, and the in-cylinder combustion characteristics and engine performance of diesel engines are compared on 50%, 75%, and 100% loads. Finally, this work makes a comprehensive comparison of the three cases and provides suggestions for case selection according to different application conditions. The conclusions can be obtained as follows:

1. By adjusting the intake pressure and fuel injection timing, the engine power under 10 kPa and 25 kPa EBP conditions is maintained at the optimum value. Engine power, fuel consumption, and HRR all increase with load increase. The change in EBP has little effect on the optimal engine power, and because the power is similar, the HRR does not change significantly with different EBPs. The EBP affects the fuel consumption rate by affecting both the in-cylinder temperature and the fresh air intake volume. Whether it increases or decreases the fuel consumption rate mainly depends on which influence mechanism plays a decisive role.
2. Advanced injection timing and increased residual exhaust gas will advance CA5, and the increase in EBP will cause the initial temperature in the cylinder to rise, which also advances CA5. The trend of CA50 is similar to that of CA5, but there are some differences, mainly because CA5, which reflects ignition, is mainly controlled by injection timing and premixed combustion, while CA50 is mainly determined by diffusion combustion. Earlier injection timing and lighter load will lead to insufficient fuel–air mixing in the late combustion period, prolonging the combustion duration.
3. The increase in EBP and the decrease in valve overlap angle will lead to the increase in residual exhaust gas and initial temperature, which will promote the mixing of fuel and air during the ignition delay period, accelerate the initial combustion rate, and increase the maximum pressure rise rate and peak pressure. The position of the ignition time relative to 0° CA becomes an important factor for the peak pressure by affecting the combustion speed during the rapid combustion period and the cylinder volume during the rapid heat release stage.
4. Case 1 is applicable to diesel engines with increased EBP caused by EGR and after-treatment systems, while Case 3 is applicable to diesel engines with increased EBP caused by underwater exhaust. Case 2 is suitable for diesel engines with an EBP higher than one atmosphere, while the injection pressure and intensification are limited by technology and materials.

**Author Contributions:** Conceptualization, L.H. and L.L.; methodology, L.H.; software, R.L.; validation, L.H., R.L. and Y.W.; formal analysis, L.H.; investigation, J.L.; resources, L.H.; data curation, J.L.; writing—original draft preparation, J.L.; writing—review and editing, Y.W. and L.L.; visualization, J.L.; supervision, L.L.; project administration, J.L.; funding acquisition, L.H. All authors have read and agreed to the published version of the manuscript.

**Funding:** This research was funded by Natural Science Foundation for Distinguished Young Scholars of Heilongjiang Province, grant number JQ2020E005.

**Institutional Review Board Statement:** Not applicable.

**Informed Consent Statement:** Not applicable.

**Data Availability Statement:** Not applicable.

**Conflicts of Interest:** The authors declare no conflict of interest.

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
