# Peer review of "Experimental Investigation on Combustion and Performance of Diesel Engine under High Exhaust Back Pressure"

_machines, doi:10.3390/machines10100919_

Round 1

Reviewer 1 Report

This paper studied the combustion process and engine performance of a marine engine with different exhaust back pressure. The paper can be considered to be published after addressing the below comments:

1.       Please remove the first paragraph of Introduction section.

2.       Please carefully proofread the paper. There are a couple of typos and grammatical errors.

3.       Table 2. Please clarify why the two back pressures were selected for this study.

4.       Table 2. As the intake pressure, injection timings and pressures were changed for different cases, it might be very difficult to clarify the impact of single variable when explaining the results (e.g. combustion process, engine performance etc). Besides, please also clarify why these injection pressures/timings were selected?

5.       It is not clear the mixture lambda in the experiments. Please clarify.

6.       Please clarify how was the load defined. IMEP or BMEP or others? Similarly, please clarify if the engine power the indicated or braked power.

Reviewer 2 Report

The study concerns the combustion and performance of Diesel engines used in ships.

There is probably a fragment of the template in the text. Please remove the fagment of the template: Line 27-35.

In the experimental setup part, the diagram should be completed (Figure 1) - how the alternator is loaded - how was the alternator / engine load was determined - how the load level was determined. It should also be supplemented in the description.

It seems that conducting trials for a larger number of EGR values ​​would be more interesting due to the analysis carried out.

In the analysis, the EGR recirculation rate could be also estimated

Additionaly there are a few editorial flaws in the text in the form of lack of spaces between words / figures.
